# Upcycling Different Particle Sizes and Contents of Pine Branches into Particleboard

**DOI:** 10.3390/polym14214559

**Published:** 2022-10-27

**Authors:** Anita Wronka, Grzegorz Kowaluk

**Affiliations:** Institute of Wood Sciences and Furniture, Warsaw University of Life Sciences—SGGW, Nowoursynowska St. 159, 02-776 Warsaw, Poland

**Keywords:** wood, upcycling, particleboard, mechanical properties, bulk density, physical properties, carbon storage, wood branches

## Abstract

A growing world population means that demand for wood-based materials such as particleboard is constantly increasing. In recent years, wood prices have reached record highs, so a good alternative can be the utilization of branches, which can reduce the cost of raw materials for particleboard production. The goal of the study was to confirm the feasibility of using an alternative raw material in the form of *Pinus sylvestris* L. pine branches for the production of three-layer particleboard. Characterization of the alternative raw material was also carried out, and the bulk density was determined. As part of the research, six variants of particleboard, 0%, 5%, 10%, 25%, and 50%, *w*/*w*, and two variants where the first one had the face layer made of branch particles and the core layer made of industrial particles, and the reverse variant (all produced panels were three-layer) were produced and then their physical and mechanical properties were studied. The results show that even if the bulk density of branch particles is significantly higher than industrial material, the internal bond and water absorption rises as branch particle content increases. In the case of bending strength and modulus of elasticity, these were decreased with a branch particle content increase. The conducted tests confirmed the possibility of using the raw material, which was usually used as fuel or mulch, to produce particleboards even in 50% content. The present solution also contributes to the positive phenomenon of carbon storage, due to incorporating the branches’ biomass into panels rather than burning it. Further research should be focused on the modification of particle production from branches to obtain lower bulk density and to reach fraction shares closer to industrial particles. Furthermore, the chemical characterization of the pine branch particles (cellulose and lignin content, extractives content, pH value) would provide valuable data about this potential alternative raw material.

## 1. Introduction

In times of scarcity of wood, it is extremely important to make the best possible use of once harvested raw material. Sometimes this is not enough, so it is necessary to look for new alternative raw materials. Branches are very often not widely used in the wood industry due to their small diameter, irregular shape and size, and a high proportion of bark, among other reasons. In pine farms, pruning is carried out every few years, removing branches to obtain as much wood as possible in a short period. In pine forests, on the other hand, unwanted branches are also regularly removed to obtain better wood quality. In addition, stands that obstruct the growth of more mature trees are also removed. Each of these operations produces by-products that can be used as fuel, compost, or ballast for trees [1], or left in the forest during harvesting [2], so using this waste can have positive benefits for the environment and the production of wood-based panels. A growing world population means that demand for wood-based materials such as particleboard is constantly increasing. The worldwide amount of particleboards produced in 2020 was 96.01 million m^3^ [3]. In recent years, wood prices have reached record highs, so a good alternative is to use branches, which can reduce the cost of raw materials for particleboard production. This solution gives hope to wood materials producers to increase the profitability of production, but it also allows carbon dioxide to remain bound in the atmosphere for longer. This approach fits well with the general direction of EU policy concerning Carbon Capture and Storage [4]. The Scots pine tree accounts for 67% of Poland’s forest area. It can be found in almost all environmental conditions, which proves its easy availability [5]. Branches, as well as high-grade wood, need to be mechanically processed. The research shows that the chipping of branches produces smaller particles than particles extracted from pine logs. The amount of dust was also higher in the particles produced from the branch material. The significantly higher amount of fine particles in pine particles may be related to the presence of needles. The smaller particles used in particleboard production may contribute to lower strength parameters [6].

Research conducted so far does not confirm the high popularity of using branches in the aspect of wood-based panel production, so it is difficult to find examples strictly related only to pine branches. In Iran, due to the scarcity of conventional wood raw materials, agricultural waste started to be of interest in the context of obtaining raw materials for particleboard production, thus date palm (*Phoenix dactylifera*) branches were used, which were considered to be easily available. Research has confirmed that date palms can be used as a substitute for conventional raw material and that the boards produced can serve as a sound and heat insulating material [7]. A similar reason—the search for alternative raw materials for particleboard—has been a starting point to produce the particleboards from sorghum bagasse bonded with maleic acid [8] and agro-industrial residues (cassava stem, sengon wood waste, and rice husk) and different contents of natural rubber latex adhesive [9]. Both papers confirmed the usefulness of the tested alternative raw materials in particleboard production.

Pine branches have so far been used in particleboard production as a finishing element on the board. This application was intended to explore the decorative possibilities and also to produce low-emission particleboard covered with pine branches. The pine branches were cut into thin slices so that the annual rings are visible. The slices have been overlapped on both sides of the board and there are many possible ways of arranging them to create interesting patterns. Thanks to the pressing of the pine slices, the surface of the board is smooth, allowing the sanding process to be omitted and the board to be varnished more quickly. For the middle layer of the particleboard, the waste from wood production was used, which additionally increases the ecological value of the board. Such boards can be used as decorative furniture fronts or as decorative panels for construction elements [10].

Another example of branches that are used in the wood industry is those of fruit trees grown in Greece and evergreen deciduous shrubs. Every year, 415,000 tonnes of green woody biomass from apple, apricot, peach, pear, and cherry trees are left in the fields or burnt. Studies have confirmed the possibility of using wood particles from fruit trees and evergreen shrubs in combination with Greek fir wood to produce particleboard that meets the requirements for EN 312:2010 class P4 (load-bearing boards for use in dry conditions). The content of the alternative raw material was above 50% [11].

Research has been conducted into the possibility of using wood particles from apple tree branches, which come from the annual maintenance of these trees. Most often, these branches are used as fuel, or if left in the field, they contribute to the development of diseases and fungi. Various variants of panels were produced as part of the tests, of which only variants that had less than 50% branch particles in their composition met the standards. A higher proportion of apple branches lowered the parameters of the panels produced. This may be due to the higher bulk density of apple wood (510–600 kg m^−3^) [12]. To date, other unusual raw materials used in particleboard production include raspberry stems (*Rubus Idaeus* L.) [13], sunflowers [14], bamboo branches and wastes [15], particles of the trunk and branches of the bhadi tree (*Lannea coromandelica*) [16], grape trees pruning residues [17,18], dry branches of *Araucaria angustifolia* (*Bertol*.) [19], particles of *Phoenix Dactylifera*-L (Date Palm) [20], or chilli pepper steams [21].

Research confirms that tree branches have so far been successfully used to produce lightweight structural panels, whereas from a chemical point of view, they can also be used as biofuel, for example [22].

The present research was aimed at proving that it is possible to use pine branches for the production of particleboard, the properties of which meet standard requirements for use in the furniture industry. The research also included the characterization of the wood particles obtained by grinding the branches, and their bulk density and geometry were determined.

## 2. Materials and Methods

### 2.1. Materials

Pine (*Pinus sylvestris* L.) branches that were left as waste in the forest during felling were used in the study. The largest diameter of the collected branches was about 40 mm, whereas the lowest was about 10 mm. The collected branches were dried using a chamber dryer, where the temperature was 70 °C, and the drying resulted in a branch equivalent moisture content of approximately 10–12%. The dried branches were pre-crushed on a saw blade into 50 mm long chips, and the next step was to mill these chips using a laboratory hammer mill. As a reference material, the industrial particles consisting of over 95% of *Pinus sylvestris* L. have been taken from one of the commercial 3-layer particleboard production lines located in Poland.

An air gun was used to spray the glue over the particles, using a commercial urea-formaldehyde (UF) resin Silekol S-123 (Silekol Sp. z o.o., Kędzierzyn-Koźle, Poland) with a molar ratio of 0.89 and solid content of 66.5%. The resination was as follows: 12% for face layer particles and 10% for core layer particles of dry resin content referred to dry particles (*w*/*w*), where 2.0% of water solution of ammonium nitrate hardener was applied, and both were calculated for dry resin content. No hydrophobic agents were added.

### 2.2. Wood Raw Material Upcycling and Characterization

The chips made of pine branches were re-milled on a laboratory knife mill (laboratory prototype delivered by Research and Development Centre for Wood-Based Panels Sp. z o. o. in Czarna Woda, Poland), equipped with three knives, two contra-knives, and a 6 × 12 mm^−2^ mesh. Using the volumetric method, the bulk density of the particles obtained was tested. The measurement was repeated five times for each fraction. The particles obtained were sorted into a face layer (0.5 mm and 1 mm sieves) and a core layer (8 mm and 2 mm sieves). This procedure allowed the elimination of oversized particles. The fractions of particles were tested with an IMAL (Imal s.r.l., San Damaso (MO), Italy) vibrating laboratory sorter with seven sieves. The selected sieve sizes were 8, 4, 2, 1, 0.5, 0.25, and <0.25 mm. The amount of tested material for each fraction was about 100 g, and the set time of continuous vibrating was 15 min. As many as five repetitions were done for every tested material.

A brief characterization of *Pinus sylvestris* L. bark mechanical parameters (internal bond strength according to [23]), as well as the density was also conducted. The dimensions of the bark samples taken (as in Figure 1) were about 30 mm × 30 mm. As many as 25 samples have been used for every test. The bark content (*w*/*w*) for 50 mm long branch sections was also established. The bark was manually separated from 50 samples.

### 2.3. Elaboration of Composites

Three-layer particleboard with different branch particle contents and in three different combinations was produced from pine branch particles. The particles used were dried to a moisture content of 5%. All particleboard variants produced had a density of 670 kg m^−3^, 32% (*w*/*w*) face layers content, and a nominal thickness of 16 mm. The tests produced reference particleboards and particleboards with branch particle content sequentially: 0%, 5%, 10%, 25%, 50%, and 100% by weight, as well as particleboards in which the face layer was produced as in the reference particleboard (industrial particles) but 100% of branches particles in the core layer (called here 100 cl) and the opposite structure (100% branch particles in face and 100% of industrial particles in core) were called 100 fl. The manually formed mats of particleboards were pressed on a hydraulic press (ZUP-NYSA PH-1P125) at a maximum unit pressure of 2.5 MPa, a temperature of 200 °C, and a time factor of 20 s per one mm of nominal panel thickness. The conditioning of the panels before the tests took place at 20 °C and 65% humidity until a constant mass was obtained. The entire process of an attempt of upcycling the wood raw material from branches is presented in Figure 2. The composition of produced panels is presented in Table 1.

### 2.4. Characterization of the Elaborated PANELS

The test specimens were cut on a saw blade as required by European standards EN-326-2 [24] and EN-326-1 [25]. The modulus of rupture (MOR) and elasticity (MOE) were determined according to EN 310 [26], and the internal bond (IB) was determined according to EN 319 [23]. All the mechanical properties were examined with an INSTRON 3369 (Instron, Norwood, MA, USA) laboratory-testing machine, and, whenever applicable, the results were referred to as standards [27]. Board density was determined according to EN 323 [28], thickness swelling (TS), and water absorption (WA) due to EN 317 [29]. The screw withdrawal resistance (SWR) was measured according to [30]. The density profiles of the tested panels were measured on a GreCon DAX 5000 device (Fagus-GreCon Greten GmbH and Co. KG, Alfeld/Hannover, Germany).

### 2.5. Statistical Analysis

Analysis of variance (ANOVA) and t-tests calculations were used to test (α = 0.05) for significant differences between factors and levels, and where appropriate, using IBM SPSS statistic base (IBM, SPSS 20, Armonk, NY, USA). A comparison of the means was performed when the ANOVA indicated a significant difference by employing the Duncan test. The statistically significant differences in achieved results are given in the Results and Discussion paragraph whenever the data were evaluated.

## 3. Results and Discussion

### 3.1. Materials Characterization

The measured density of mature bark samples was 275 kg m^−3^ ± 21 kg m^−3^, whereas the density of juvenile pine bark was 413 kg m^−3^ ± 18 kg m^−3^. The measured internal bond of mature bark samples was 0.07 N mm^−2^ ± 0.01 N mm^−2^, since, in the case of juvenile bark, the internal bond values were almost zero. These samples were delaminated during sample preparation and fixed in a testing machine. The measured bark content (*w*/*w*) on the branch sections was about 11%. According to the literature [31], the bark content in the mature pine log is 6.7%.

The results of the fraction share analysis of particles used in the research have been displayed in Figure 3. As it can be seen, in the case of face layer particles, a higher amount of smaller particles has been produced from pine branches. This could be caused by a higher amount of bark content in pine branches. In the case of core layer particles, a large amount of coarse particles was made from pine branches. This was especially visible for fractions 4 mm in size.

The bulk density of the alternative raw material of pine (*Pinus sylvestris* L.) branch particles (Figure 4) was higher for both the outer layer (by 54 kg m^−3^) and the inner layer (by 63 kg m^−3^). This higher bulk density of branch particles can be influenced by the high content of bark, mentioned above. However, the bark density is lower than that of pine wood density, but the weak bark mechanical properties, including internal bond, lead to the production of a high amount of fine particles, including dust, which contributes to high bulk density. The higher bulk density of the raw material may adversely affect the strength values of the panels produced, as was the case with panels made from branches from pruning fruit trees [12]. Particles from branch shredding have a different geometry and a higher bark content, which also reduces strength parameters.

### 3.2. Modulus of Rupture and Modulus of Elasticity

In the accompanying graph (Figure 5), it can be seen that as the pine branch particle content increases, the MOR and MOE values decrease. For MOR, the variant with an alternative particle content of 10% showed almost comparable results to the reference panels. The largest statistically significant differences were shown between the reference panels and the panels with an alternative raw material content of 100%. Furthermore, statistically significant differences have been found for panels 0 and 10, 25, 50, 100 cl, and 100 fl. The average values of MOR of the panels 100 against 100 cl and 100 fl were also statistically and significantly different. All the variants produced met the requirements of the EN standard.

The MOE values obtained are lower, with a difference of 1320 N mm^−2^ between the highest and lowest parameters, so all variants, in this case, met the applicable standard. The lowest values were characterized by boards made entirely of alternative particles, where the MOE was 2235 N mm^−2^. Concerning the statistical significance of the mean values of MOE, the relations found in the case of MOR can be applied.

Due to the nature of the alternative material, which is anatomically very similar to the pine wood commonly used for the production of wood-based composites, better strength parameters were expected, but due to the smaller diameters of the pine branches, they have lower strength and more bark, which has a lower density and therefore adversely affects the strength parameters. Branches with very small diameters may be too small after shredding, resulting in a large amount of dust, which is not desirable for the production of wood-based materials.

Similar relationships regarding MOE in their study were obtained by researchers investigating the strength of particleboards made with the addition of particles derived from sequoia branches [32].

### 3.3. Internal Bond and Screw Withdrawal Resistance

The results of the internal bond were presented in Figure 6. As the proportion of particles derived from pine branches increased, a favorable effect on the resistance values for internal bonds was noted. The results for the reference boards were 0.77 N mm^−2^ whereas the results for the boards made from 100% branch particles were 1.48 N mm^−2^. The best results were obtained for the variant in which the core layer of the three-layer particleboard was made from 100% branch particles and the face layer was made from particles used as standard in the industry. The values obtained for this variant were at 1.69 N mm^−2^ and were thus more than double that of the variant where the middle layer was made industrially and the outer layer was made entirely of particles from branches. In addition, all variants meet the requirements of the standard, and statistical analyses for the alternative raw material used show no statistically significant differences between the average IB values of 5 and 10% and 25 and 50%, as well as for 100 and 100 cl. Alamsyah et al. [33] in their study indicate that IB results can be influenced by the mixing of particles with glue, sheet molding, and pressing and that the higher bulk density of the alternative raw material translates into higher IB results. In another study, the researchers used mulberry branches for particleboard, and the IB study obtained the following results sequentially for variants depending on particles size: 0.25–1 mm—1.43 N mm^−2^, 1–2 mm—1.54 N mm^−2^, and 2–4 mm—2.30 N mm^−2^, at a particleboard density of 800–830 kg m^−3^ [34].

The only statistically significant differences among the average values of screw withdrawal resistance have been found between samples 100 and 100 cl (Figure 7). However, it is worth adding that the achieved values of SWR, between 156 N mm and 177 N mm were significantly higher than the SWR values found by [12] for the panels of the same purposes.

### 3.4. Thickness Swelling and Water Absorption

The accompanying graph (Figure 8) shows the results of swelling per thickness of the tested composites after 2 h and 24 h of soaking in water. When considering the two additional custom variants, after 2 h of soaking, the lowest TS was obtained for the variant in which the core layer was made entirely of alternative raw material, which was 22%. The highest value was obtained for the variant in which only the face layers were made of 100% alternative raw material; the TS was 33%. The rationale for this result can be found in the higher bark content of the alternative raw material used, which is more porous and has a lower specific density, which means it can absorb more water [35].

Analyzing the standard variants with the content of particles derived from branches from 5 to 100%, the lowest TS value after 2 h was recorded for the board made from 100% alternative raw material; TS was 23%, with the highest value after 2 h recorded for the 25% variant. With a longer soaking time, TS increases, and the proportion of dimensional change after 24 h is similar to that measured after 2 h.

WA results of the different variants of the manufactured branch particleboards are shown in Figure 9. After 2 h, water absorption was lowest for the reference variant—72%—whereas the highest values were recorded for variants 25 and 100% (80%). After 24 h, the discrepancies between the variants were slightly greater; still, the lowest WA value was characterized by the reference panels, where the WA was 82%. The highest WA value was 93% and was recorded for the variant made from 100% alternative material.

### 3.5. Density and Density Profiles

The obtained results of the density profile measurement are shown in Figure 10. The highest density in the panel was found in the outer layer, where 930 kg m^−3^ was recorded for industrial particles at a depth of about 1.5 mm measured from the surface. The lowest density in the outer layer was recorded for the variant that consisted of 10% particles from pine branches, the maximum recorded value, and in this case, was 900 kg m^−3^, and was also at a depth of 1.5 mm. The lowest density of the entire panel was recorded for the variant that consisted of 100% pine branches. That lowest density was 520 kg m^−3^ at a depth of 7 mm. All the produced variants of panels have the same density of approx. 670 kg m^−3^

Higher density of the face layers of composites very often has a positive effect on strength parameters such as MOE and MOR. It is this part of the board that is responsible for transferring compressive and tensile stresses during bending. Due to the great similarity of the alternative material to the conventionally used raw material, it is difficult to notice significant differences in the density profile graph between the variants.

## 4. Conclusions

According to the conducted research and the analysis of the achieved results, the following conclusions and observations can be drawn:It has been confirmed that particles of pine (*Pinus sylvestris* L.) branches can be successfully upcycled and used in the production of three-layer particleboard. This profitable opportunity contributes to the policy of carbon capture and storage.The bulk density of particles derived from pine branches is characterized by a higher bulk density in both the face and core layer particles in comparison with industrial particles.The content of pine branch particles was found to have a significant effect on the orthogonal tensile strength (IB): increasing the proportion of pine branch particles results in an increase in IB strength. Each of the produced variants meets the requirements of EN 312:2010 for P2-type panels.There is no significant influence of the pine branch particles share in particleboards on their screw withdrawal resistance; however, the highest SWR has been found for the panel made of 100% of pine branch particles in the core layer and 100% of industrial particles in the face layers.The highest thickness swelling of the tested panels has been found for those made of 25% of pine branch particles, whereas TS for panels made of 100% of pine branch particles in the core layer and 100% of industrial particles in the face layers were found in the lowest.The water absorption test showed increasing dynamics in soaking time as the proportion of particles from pine branches increased. In addition, the general rule is that the water absorption rises with the pine branch particles’ content.The density of the face layers of the produced composites is comparable for the different variants; however, the slight tendency of the higher densification of the face layers has been found with the increase of the pine branch particles share.Since several features of produced particleboards have been significantly influenced by bulk density and size of branch particles, further research should be focused on the modification of particle production from branches to obtain lower bulk density and to reach fraction shares closer to industrial particles. Furthermore, the chemical characterization of the pine branch particles (cellulose and lignin content, extractives content, pH value) would provide valuable data about this potential alternative raw material.

## Figures and Tables

**Figure 1 polymers-14-04559-f001:**
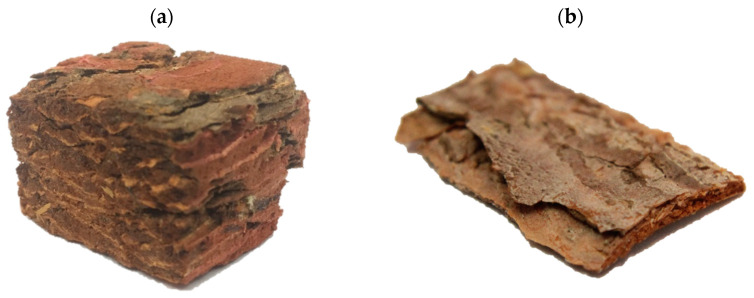
The samples of the *Pinus sylvestris* L. bark. (**a**) mature and (**b**) juvenile were taken for investigation.

**Figure 2 polymers-14-04559-f002:**
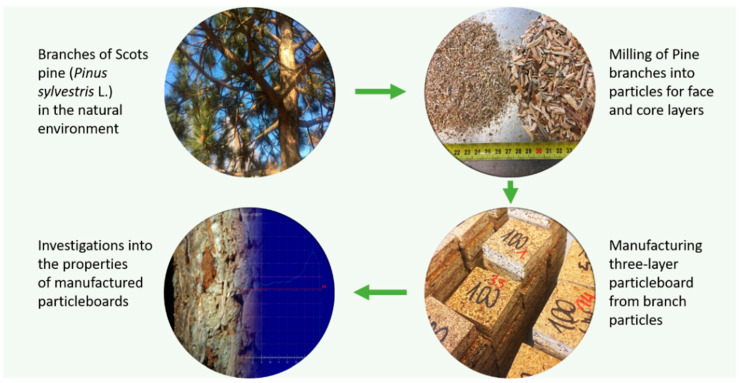
The general production process of three-layer particleboard from pine branches.

**Figure 3 polymers-14-04559-f003:**
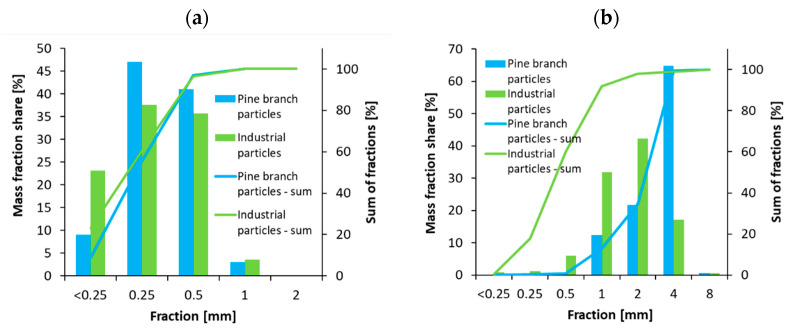
The fraction analysis results of the face (**a**) and core (**b**) layer particles.

**Figure 4 polymers-14-04559-f004:**
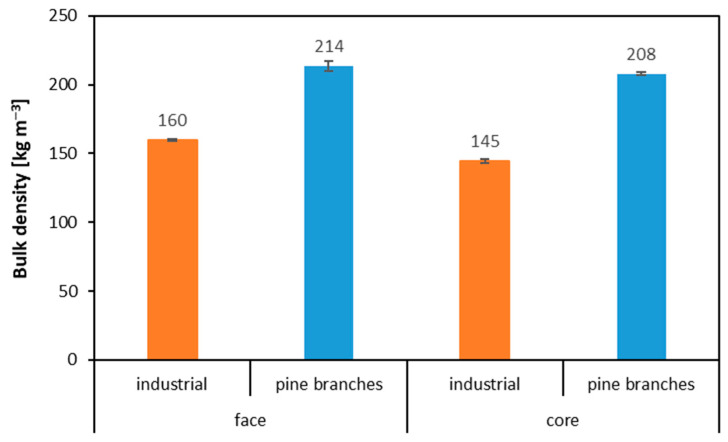
Bulk density of particles from pine branches used in the manufacture of three-layer particleboard.

**Figure 5 polymers-14-04559-f005:**
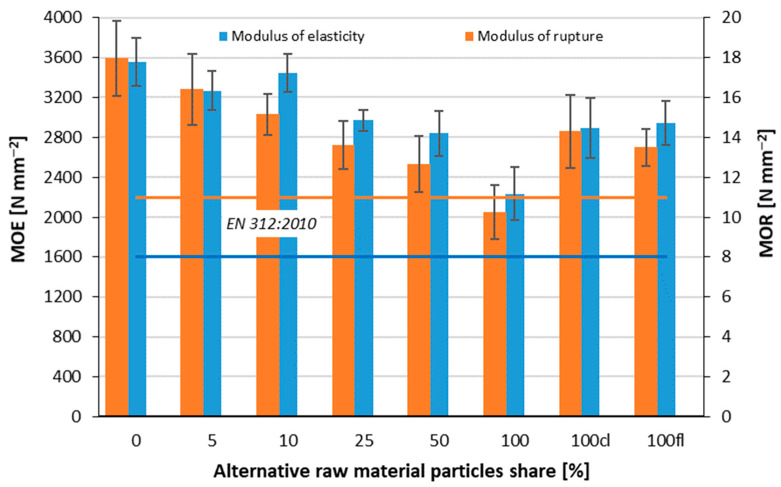
Modulus of rupture and modulus of elasticity of tested composites.

**Figure 6 polymers-14-04559-f006:**
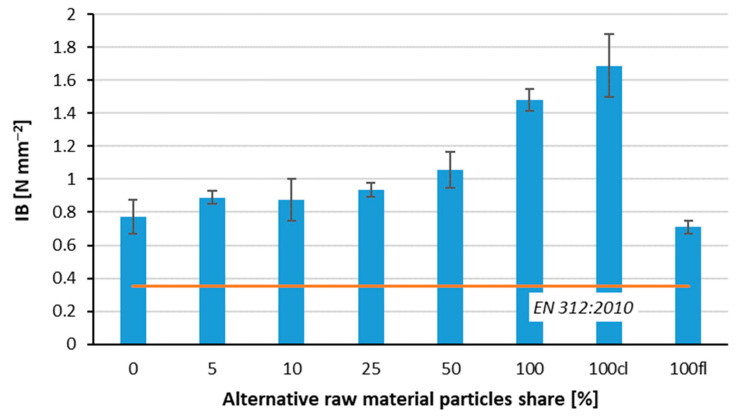
Internal bond of tested composites.

**Figure 7 polymers-14-04559-f007:**
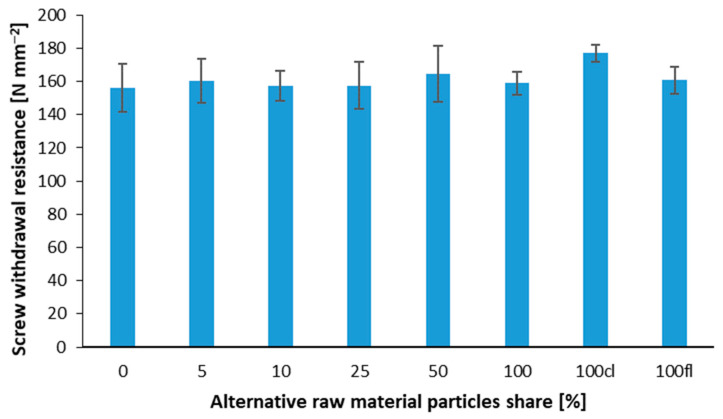
Screw withdrawal resistance of the panels with various branch particles share.

**Figure 8 polymers-14-04559-f008:**
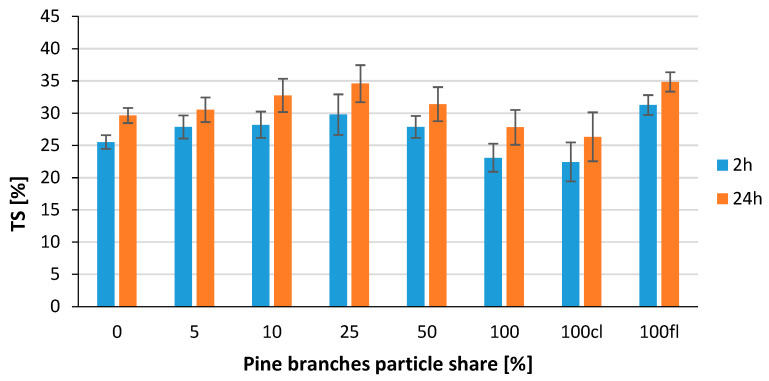
Thickness swelling of tested composites.

**Figure 9 polymers-14-04559-f009:**
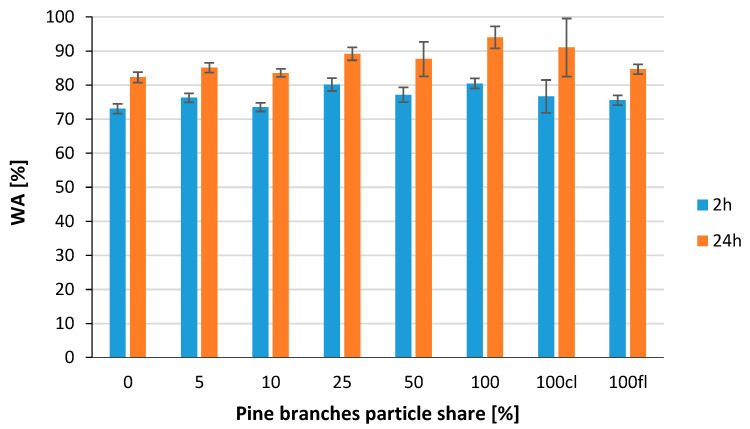
Water absorption of tested composites.

**Figure 10 polymers-14-04559-f010:**
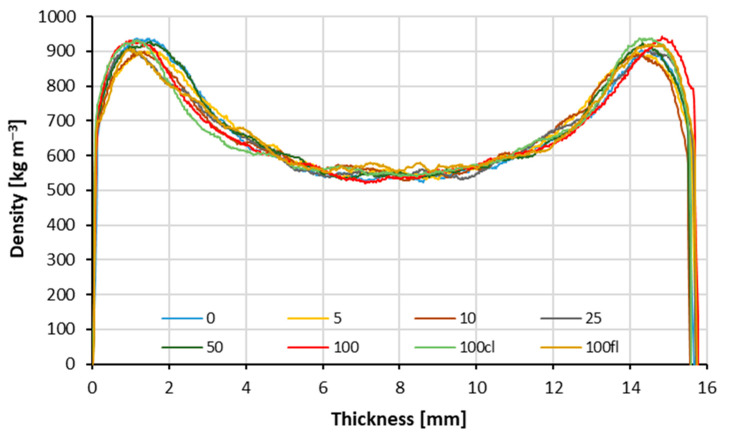
Density profiles of tested composites.

**Table 1 polymers-14-04559-t001:** Compositions of elaborated particleboards.

Panel Name	Pine Branches Particles Content [% by Weight]		Industrial Particles Content [% by Weight]	
	Face Layer	Core Layer	Face Layer	Core Layer
0	0	0	100	100
5	5	5	95	95
10	10	10	90	90
25	25	25	75	75
50	50	50	50	50
100	100	100	0	0
100 cl	0	100	100	0
100 fl	100	0	0	100

## Data Availability

The data presented in this study are available on request from the corresponding author.

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
