# Peer review of "Upcycling Different Particle Sizes and Contents of Pine Branches into Particleboard"

_polymers, 2022, doi:10.3390/polym14214559_

Round 1

Reviewer 1 Report

The manuscript is well written, only some minor changes are needed before publication.

The Title: Please consider changing the Title to "Pine Branches as an Alternative Raw Material..."

The Abstract: Please be more clear in lines 11-13, which is a single layer PB, which 3 layers, etc.

Please be more specific about the results of your research and about novelty at the end of the abstract. Also please explain how your solution contributes to carbon storage.

The Introduction: Please add some numbers about amounts of branches, which can be used for PB production.

Lines 21-46: This part is too long, some parts are mentioned repeatedly.

I suggest starting the Introduction with the research of alternative raw materials such as agricultural biomass (wood and non-wood) and recycled wood waste and by-products in PB production due to increased global demand for wood raw materials, growing environmental concerns, and legislative regulations related to cascading wood use and prioritizing value-added applications of wood resources.

Line 33:  In 2020, the production quantity of particleboard reached 96.01 million m3 worldwide.

Materials and methods:

Please consider starting with branches, then milling, then gluing of particles.

In part 2.3 Table would be easier to see the complete methodology of sample types.

The Results and discussion part are well written. Please add limitations of your research and implications for further research. Also please add novelty to your research.

Author Response

Attached file

Reviewer 2 Report

General comments:

The article titled “Pine Branches Upcycled to an Alternative Raw Material in Particleboard Production” presents insightful knowledge on the upcycling of pine branches into particleboard.

My general comments are:

·       The originality of the paper is average, with a similarity index of 19-21% to the published work.

·       The study is well documented, being in line with the guidelines for the authors, imposed by the journal.

·       The abstract is well presented, according to this research.

·       The state-of-the-art written in the introduction of the paper is well documented and focused on the actual research direction in the field.

·       The introduction presents properly the aim of the study.

·       The experimental procedure is justified but less comprehensive, specifically on the adhesives properties.

·       The results are presented in a concise manner, but low resolutions figures.

·       Furthermore, compared to references the results obtained are significant. The discussion section is clear, understandable and in accordance with the results obtained.

·       The conclusion is well written to conclude the results.

·       The reference format should be revised according to the journal requirements.

However, the article is lack of major scientific soundness. Thus, the article can be considered after addressing the below comments in the revision (major revision).

Detail Comments:

1.     Title: The title seems correct, but it needs a bit revision. Please revise to “Upcycling Different Contents of Pine Branches into Alternative Particleboard”. Authors need to write the title specifically.

2.     Abstract:

a.      Page 1. Line 9. Authors should write the background of the study. Why do you need to do this research. Revise this.

b.     Page 1. Line 14-15. You have to write a brief results of your study and discuss it in 2-3 sentences. Please expand the abstract!

c.      Page 1. Line 16. Please write your conclusion and suggestion for future research.

d.     Page 1. Line 18. The keywords are similar to the published works. Please re-write the keywords into more unique words.

 3.       Introduction:

a.      Page 1. Please extend the introduction by including previous studies on Eco-Friendly Particleboard. I believe the following articles could be used as references to improve the manuscript:

·       Sutiawan, J., Hadi, Y. S., Nawawi, D. S., Abdillah, I. B., Zulfiana, D., Lubis, M. A. R., Nugroho, S., Astuti, D., Zhao, Z., Handayani, M., Lisak, G., Kusumah, S. S., & Hermawan, D. (2022). The properties of particleboard composites made from three sorghum (Sorghum bicolor) accessions using maleic acid adhesive. Chemosphere, 290(December), 133163. https://doi.org/10.1016/j.chemosphere.2021.133163

·       Hidayat, W., Aprilliana, N., Asmara, S., Bakri, S., Hidayati, S., Banuwa, I. S., Lubis, M. A. R., & Iswanto, A. H. (2022). Performance of eco-friendly particleboard from agro-industrial residues bonded with formaldehyde-free natural rubber latex adhesive for interior applications. Polymer Composites, 43(4), 2222–2233. https://doi.org/10.1002/pc.26535

·

b.     Page 2. Line 92 and Figure 1. Please rearrange “The entire process of an attempt of upcycling the wood raw material from branches is presented in Figure 1” into Methods section.

4.        Material and methods:

a.      Page 3. Section 2.1. Line 113-117. Please re-write/paraphrase the sentences. Most of the sentences are similar to the published works.

b.     Page 3. Section 2.2. Line 120-131. Same in here. Please re-write/paraphrase the sentences. Most of the sentences are similar to the published works.

c.      Page 4. Section 2.3. Line 142-155. Also in here. Please re-write/paraphrase the sentences. Most of the sentences are similar to the published works.

d.     Page 4. Section 2.4. Line 158-167. Please re-write/paraphrase the sentences. Most of the sentences are similar to the published works.

e.      Page 5. Section 2.5. Line 169-174. Please re-write/paraphrase the sentences. Most of the sentences are similar to the published works.

f.      Auhtors need to do chemical characterization of the pine branches particles such cellulose content, lignin content, extractive content, pH value, etc. It is important properties to investigate the potential of pine branches as raw material for particleboard.

5.       Results and discussion:

a.      Please add the results of chemical analysis of pine branches particles and their relation to the properties of the particleboard.

b.     Overall, the results are well discussed. P;lease revise all the figures with higher resolution of 1200 DPI.

6.     Conclusions

a.      The conclusion is well written in answering the objectives of the study.

Author Response

Attached file

Round 2

Reviewer 2 Report

The manuscript has been revised accordingly and it is suitable for publication in Polymers.

After reading the manuscript again, please revise the title to "Upcycling Different Particle Sizes and Contents of Pine Branches into Particleboard"

The rest are good.

Author Response

Dear Reviewer,
the title of the manuscript has been changed according to your suggestion.
Seems like now is more precise.
Thank you for that remark.

Regards!